# Adaptation of retired older adult return migrants in their place of origin in the Yogyakarta Special Region, Indonesia

Mita Noveria 1,2, Umi Listyaningsih3,4,5, Agus Joko Pitoyo3,4,5*, Aswatini Anaf2

1 Doctoral Program of Population, Graduate School, Gadjah Mada University, Yogyakarta Special Region, Indonesia, 2 Research Centre for Population, National Research and Innovation Agency, Jakarta, Indonesia, 3 Faculty of Geography, Gadjah Mada University, Yogyakarta Special Region, Indonesia, 4 Graduate School, Gadjah Mada University, Yogyakarta Special Region, Indonesia, 5 Center for Population and Policy Studies, Gadjah Mada University, Yogyakarta Special Region, Indonesia

* aguspit@ugm.ac.id

## Abstract

Migrants must adapt to the social life of their destination, including those who return to their place of origin. One such destination for older adult return migrants in Indonesia is the Yogyakarta Special Region (*Daerah Istimewa Yogyakarta DIY*), located in the centre of Java island. The majority of residents in this province, which is under the jurisdiction of the Sultanate of Yogyakarta, are of Javanese ethnicity and continue to uphold traditional Javanese cultural values in their daily social interactions. This study examines the adaptation strategies employed by retired older adults who return to their place of origin, defined as either their place of birth or previous residence, as their primary DIY location. Using a qualitative method, data were collected through open-ended interviews with 27 retired older adult return migrants, selected through snowball sampling. The findings indicate that these return migrants primarily rely on Javanese cultural values to navigate social reintegration, regardless of the length of time they lived outside the DIY region. Embracing these values facilitates smoother adaptation to the local social environment, where such traditions remain strong. This strategy could apply to migrants of other ethnicities who internalize their cultural values, some of which may share similarities with Javanese values, when migrating to a new destination or returning to their place of origin.

## 1 Introduction

Migration is a longstanding phenomenon that occurs throughout human life across different generations. The activity is influenced by life-course events, such as the completion of schooling, entry into the labour force, marriage, childbirth, divorce, children leaving parental homes due to adulthood, and retirement [1,2]. Recent studies on migration intertwine the intention and desire to move with the life-course

**Data availability statement:** The data can be accessed at the following link: https://hdl.handle.net/20.500.12690/RIN/QINETR. However, because the research participants did not consent to public data sharing, access to the data repository is available upon request at rin_rmpi@brin.go.id. The National Scientific Repository (RIN) is managed by the National Research and Innovation Agency (BRIN), which can be accessed here: https://data.brin.go.id/.

**Funding:** The research was funded by the Institute of Social Sciences and Humanities of the National Research and Innovation Agency, Indonesia (Grant number B-577/III.7/HM.06.02/2/2023. The funders had no role in study design, data collection and analysis, decision to publish, or preparation of the manuscript.

**Competing interests:** The authors have declared that no competing interests exist.

events [3], and analyse life-course events such as leaving parental home [4] and retirement [5–8].

The motivations and driving factors behind migration exhibit age-related variation, with the highest likelihood observed among younger individuals, gradually declining with increasing age [3,9]. Therefore, discussions on migration are commonly centred around the younger age groups [10]. The migration of young individuals, particularly those relocating from non-metropolitan to metropolitan areas, is driven by aspirations for educational advancement, improved job prospects, career enhancement opportunities [11], and the pursuit of economic independence [12]. Conversely, the primary motivations driving older people's migration are unrelated to economic factors [13–17]. Older adults often relocate to areas distant from workplaces and some move to their place of origin. Since the move is not constrained by employment considerations, older adults are more likely to prioritise other factors [15,18–20]. Moreover, a strong sense of place often leads older adults to return to their place of origin in later life, following periods of employment in other regions during their younger years [21].

Migration results in a shift in residential location within a specific area, ranging from changes in neighbourhoods to transitions across cities and countries [22]. Residential change also entails social transformation [23] as migrants leave their familiar environment to relocate to new areas and integrate into different communities that are likely to be socially and culturally distinct from their previous residences [24]. Residing in the destination area exposes migrants to various facets, including aspects such as pleasure and comfort, culture, religion, and patterns of social relations among community members [25]. Migrants who undergo cross-cultural transition are compelled to adapt to the new social environment [26], which may involve forming new friendships and participating in social activities [23]. Adaptation becomes more complex when migration occurs between areas with vastly different social environments. Consequently, the potential challenges migrants face in adapting to the social environment are significant considerations in their destination choices [27].

Adapting to the social environment is necessary not only for migrants relocating to new destinations but also for those returning to their place of origin. This necessity arises because migrants have lived in the destination area, where social values and norms may differ from those of their original community. Over time, migrants may have been influenced by the values and norms prevalent in the destination area and must readjust to the social environment of their origins upon returning [28].

Numerous studies on migrant adaptation have been conducted, internationally [29–39] and domestically [27,40–44]. Most of these studies did not focus on specific age groups. However, some studies have explored specific age groups, including social adaptation among migrant children in China [45], adaptation of older adult Chinese migrants in Singapore [46], and adaptation of Tibetan retirees in Chengdu, China [47]. Furthermore, [48] analysed the adaptation of older adult immigrants from six Asian countries in the United States and [49] examined the adaptation of Romanian older adult migrants in Western European countries. However, studies focusing on the adaptation of return migrants are rare, including Turkish return immigrants from Western countries [34], American return migrants from Israel [50], repatriation

of American teachers from Japan [51], and repatriation preparedness of American managers from overseas [52]. Studies on the adaptation of older adult return migrants to their place of origin are also scarce, except for Turkish older adults who migrated back home after living many years abroad, particularly in some European countries [38,53].

Despite the significant contribution of older adults to internal return migration in Indonesia, research on their adaptation experiences remains limited. While studies on older adult migration are generally scarce, a notable exception is the work of [54], who analysed the migration motives of Indonesian older adults using the 2007 Indonesia Family Life Survey (IFLS) data. However, this study did not specifically investigate the experiences of older adults returning to their place of origin. Addressing this gap, the central research question of this study examines the adaptive strategies employed by retired older adults to navigate the social environment after returning to their place of origin, with a particular emphasis on their social reintegration within their communities, especially at the neighbourhood level, following retirement [53].

This study focuses on older adults who relocated to their place of origin immediately after retirement or during the subsequent period. In contrast to the general Indonesian population, the retired older adults in this study demonstrate a higher educational attainment. National data from 2024 indicate that only 41% of the population has attained secondary or higher education [55]. The higher educational attainment among these retired older adults is likely due to the educational requirements of their former employment. Furthermore, the retired older adults in this study represent a relatively economically successful cohort, as their post-retirement financial capacity enables them to afford returning to their place of origin.

The term "place of origin" refers to either the province of birth or the province in which the older adult return migrants previously resided. In this study, the term specifically pertains to the Yogyakarta Special Region (*Daerah Istimewa Yogyakarta / DIY*). Prior to Indonesia's independence, Yogyakarta was a kingdom under the Yogyakarta Sultanate [56]. The Yogyakarta Special Region, along with Solo in Central Java Province, which before independence was under the Surakarta Sultanate, serves as a primary centre of Javanese culture [57]. Javanese cultural values, such as the imperative to achieve harmony and respect for others, are strongly maintained and practiced in people's social lives [58], although some aspects have evolved [56].

This study aims to address the gap in academic knowledge regarding the adaptation of older adult migrants in their place of origin, an area that remains understudied. Moreover, the significance of this study lies in the potential impact on the welfare of retired older adult return migrants residing in DIY. Individuals who navigate the adaptation process successfully are likely to experience a content and fulfilling old age in DIY. In contrast, individuals experiencing difficulties in adapting may face social challenges that significantly hinder their social integration, potentially driving further out-migration from their place of origin.

This paper comprises five sections arranged as follows. The first section serves as the introduction, showing the topic, context, and addressed gap. The second section is dedicated to the literature review, involving concepts, theories, and previous studies related to the adaptation of return migrants, including older adult retirees. The third section delves into the methodology, comprising the location, data sources, data collection, and data analysis. The fourth section presents the adaptation strategies of older adult retirees who return to their place of origin in DIY. Finally, the last section serves as the discussion, conclusion, and policy recommendation, summarising the main findings and the implications for older adult return migrants in DIY. It discusses the implications of their return for service provision, highlighting necessary measures to support their well-being in old age.

## 2 Literature review

Migration occurring across numerous regions worldwide leads migrants to engage in contact and interaction with individuals from diverse cultural backgrounds [59]. The movements result in cultural transformations, a phenomenon known as acculturation [29,60], which involves adaptation at group and individual levels. Although adaptation is a strategy in the acculturation process, the term is interchangeable with acculturation, adjustment, and accommodation [61].

Adaptation refers to relatively stable changes in individuals or groups to react to environmental demands [29,62,63], including different social circumstances in a new area [41]. In a study on the socio-cultural adaptation of the Chinese medical aid team in Africa, [64] referred to adaptation as the management of day-to-day life within the social context of the destination country. Moreover, in a study on cultural distance and acculturation scales, [65] referred to sociocultural adaptation as the behavioral aspects of adapting to a new culture. Individuals subjected to the process of adaptation tend to embrace public morality, such as societal obligations, goals, and behaviours [66].

Adaptation comprises numerous facets of life, including establishing new social connections, securing employment, and promoting relationships in informal groups in a novel environment, deriving benefits from the associations [29]. The process of adaptation includes navigating diverse attributes inherent to individuals with distinct social and cultural backgrounds, such as language, culinary preferences, and clothing. A study on older people migration to urban Shanghai showed the necessity for migrants to conduct adaptation in multiple aspects, including self-identity, daily activities, and social interactions [41]. Social networks and cultural maintenance are among factors in determining the success of rural migrants' adaptation in urban China [67].

The most frequently used indicators to measure adaptation are acculturation, economic success, satisfaction, and identification with the destination [37]. In a study on immigrants in Australia, [37] applied several criteria influencing migrants' adaptation, namely English fluency, emotional health, diversity of experience, interpersonal relationships, and expectations of migration activities. A study on Romanian older people migrants in western European countries found similar results that the ability to speak the host countries' language was an important factor affecting migrants' adaptation [49]. Drawing upon these findings and further supported by [30,68] and [67], limited proficiency in the host country's language consistently emerges as a substantial impediment to immigrants' socio-economic integration [29,67].

Adaptation becomes necessary following return migration due to potential transformations experienced by the individual migrant and within the socio-environmental context of their origin during the out-migration period. On one hand, after a period of residence in different locations, migrants' identities may be subjected to transformation, feeling a sense of disconnect or not fitting in with former friends, colleagues, and family members [26]. Therefore, when returning to their place of origin, these individuals have to re-adapt to the social life of the area [38,69]. On the other hand, numerous alterations in various facets of life may have transpired in the place of origin during the period of outmigration [70]. As a consequence, migrants returning home encountered challenges in adapting to changes in social dynamics since the current reality often diverged from those before they left the place of origin.

The sociocultural adaptation of migrants in their place of origin is influenced by various factors [34,53]. The factors are personality, life satisfaction concerning migration, the discrepancy between pre-migration expectations and post-return reality, views on discrimination, ethnic identity, and demographic variables. Migrants who were unsuccessful in re-adapting to the current situation of their origin country often preferred to re-migrate to their previous destinations [34].

Many studies on migrants' adaptation are related to international migration; however, the results can also be extrapolated to internal migration. The assertion is supported by a study on the integration of internal migrants in urban China, which shows similar influencing factors as observed in international migration [40]. The factors are ethnic identity, employment status, and living conditions in areas where they previously lived. Factors influencing social adaptation can similarly apply to older adult return migrants in their place of origin. [39] pointed out that the presence of social groups and individual participation in various activities were important in determining the success of immigrant adaptation in the destination area.

[62] suggested four strategies in his acculturation model: assimilation, separation, integration, and marginalization. Assimilation refers to individuals who do not desire to maintain their home culture and seek interaction with the new culture. Individuals who retain their home culture and avoid interaction with the host culture are referred to as separatists. Integration is characterized by individuals who maintain both their home and new cultures. Furthermore, marginalization is characterized by individuals who have little intention of maintaining their homes and new cultures. [28] argued that Berry's

 

model is more appropriate for individuals who move from a particular culture to a destination area. A more suitable acculturation strategy for return migrants is Sussman's model [28]. In the model referring to the repatriation of sojourners, [26] suggested four strategies of repatriation that could be considered for return migrants: subtractive, additive, affirmative, and intercultural. The subtractive strategy is described as one that intends to interact with other return migrants. The additive strategy is categorized as return migrants who seek relationships with community members from the previous host country. The affirmative strategy refers to maintaining and strengthening home culture during migration while ignoring the differences between home and new cultures. Finally, intercultural refers to holding and managing both the home and new cultures simultaneously.

The acculturation model proposed by [26] is based on cross-country migration but can be applied to internal movement, especially between regions with culturally different ethnicities, such as Indonesia. This study refers to Sussman's acculturation model in analysing the adaptation strategies of older people who return to their home areas after retirement. Referring to [64] and [65], adaptation in this study is a socio-cultural adaptation, denoting the strategies of retired older adult migrants in managing their daily lives to "fit in" with the society in their area of origin, namely the Yogyakarta Special Region.

## 3 Materials and methods

### 3.1 Data collection and analysis

This study applied a qualitative method and data were collected through in-depth interviews with retired older adults (aged 60 years and older) who migrated to their place of origin. The participants were retired older adults who were either born in DIY or not born in DIY but had lived in the province for at least one year, left during economically productive years, and returned for permanent residence after retirement. The selection of participants was conducted using the snowballing technique.

Interviews were systematically conducted, starting with participants known to the researcher. The method used was similar to [41,71], which selected the initial participant based on the recommendation of friends and colleagues. Participants were asked questions regarding strategies they carried out for re-adapting to community social life after moving back to DIY to live permanently.

The initial participant was a retired older woman whose husband is an acquaintance of the researcher. She was born in DIY, left the province upon marriage, and returned to the province of birth after retirement. The contact numbers of subsequent potential participants were acquired from previous ones with explicit permission. Subsequently, prospective participants were contacted through WhatsApp messenger or a phone call to assess their willingness to participate in the interview process. Following the affirmative expression of willingness, appointments were scheduled at the locations during convenient times. All the interviews were carried out at participants' houses and each session lasted approximately one to two hours.

Data collection, through interviews with participants by the researchers, began on May 22, 2023 and ended on August 31, 2023. Interviews were recorded using a voice recorder with the participant's permission and the recordings were transcribed into verbatim form. Subsequently, the collected data were coded according to the process, starting from open, axial, and selective coding. The data were stored and managed using the ATLAS.ti 23 programme. A comprehensive descriptive analysis was used to analyse the data according to the codes following the thematic framework.

This study received ethical approval from the Ethics Committee for Social Sciences and Humanities at the National Research and Innovation Agency of Indonesia (Approval No. 091/KE.01/SK/04/2023). The approval was authorized by Dr. Augustina Situmorang, M.A., Director of the Ethical Commission on Social Sciences and Humanities. All participants were informed about the nature and purpose of the research and provided their informed consent prior to data collection. Although the COVID-19 pandemic has largely subsided, it remains a public health concern. In accordance with

institutional and governmental guidelines, all research activities were conducted in strict adherence to COVID-19 safety protocols to ensure the health and safety of both participants and researchers.

## 3.2 Study area

Yogyakarta Special Region (*Daerah Istimewa Yogyakarta Province / DIY*), located on Java Island comprises four districts, namely Sleman, Bantul, Kulon Progo, and Gunung Kidul, and one city, namely Yogyakarta, the capital city of DIY. Yogyakarta has long been known as a city of students [72], culture, struggle, education, tourism, and a "cosy heart" [73]. The availability of diverse educational facilities serves as a significant enticement and the nature of the population contributes to the city's attractiveness. Additionally, the comparatively lower cost of living in Yogyakarta and DIY as a whole, in contrast to other urban centres, enhances the appeal.

The societal dynamics in the DIY community bear a significant imprint from the Mataram Kingdom. As members of the Javanese ethnic group, individuals experience a profound impact, shaped by the prevalent norms and values inherent in the Javanese culture. A community social life is based on some Javanese cultural values namely *gotong royong* (helping one another as community members), *guyub rukun* (a sense of togetherness that exists in the community based on harmonious living), *grapyak semanak* (friendly and easy to get along with others), *lembah manah* (humble), *ewuh pakewuh* and *pangerten* (respect the feelings of others), *andhap ashor* (respect others who are older or superior), and *tepo seliro* (tolerance) [74].

Cultural values are derived from the two principles governing interaction of people in Javanese society, namely *rukun* (harmony) and *hormat* (respect others) [75]. People should maintain the harmony manifested in behaviour to prevent conflict [71,75]. Furthermore, respect is expressed through attitudes and manners of speaking that are consistent with the status and position of others in the society [75]. By upholding these principles, communities can promote harmonious relationships and reduce the probability of social conflicts.

The main and dominant language spoken by people in DIY is Javanese. The language has several levels and its use depends on the interlocutor [76,77]. Generally, there are two levels in the Javanese language, namely *ngoko* (low level) and *krama* (high level). Both *ngoko* and *krama* consist of multiple levels determined by the position of the interlocutor's age, gender, kinship relation, and social status in society, such as economic and political power [77]. The use of *krama* Javanese is a form of respect [78] in speech and conversation with interlocutors who have a higher social and economic status. The cultivation of respect begins at a young age in a familial setting to highlight the importance of using proper language. According to Javanese cultural values, the appropriate use of language is a manifestation of *ewuh pakewuh* (meaning respect the feelings of others).

DIY is considered an ideal residential destination for older adults. The appeal lies in the prospect of a serene and comfortable lifestyle, cost-effective living, and the ready availability of essential services, drawing individuals, including older adult retirees, to settle in the province [79,80]. This is consistent with the statement of the DIY's governor, the tenth Sri Sultan Hamengku Buwono, who stated that the region's peaceful, secure, and comfortable environment, with mutual respect and appreciation among residents, irrespective of ethnicity and place of origin, renders DIY an ideal destination for older adult retirees [81].

## 4 Results

### 4.1 Participants' social demographic and economic characteristics

This study involved 27 older adults, comprising 19 men and 8 women, who had returned to DIY to reside permanently following retirement. Participants ranged in age from 60 to 72 years. Of the total sample, 24 were married, two were widowed, and one female participant was single. Most married participants lived solely with their spouses, without co-resident children. Most participants were not residing in the same location they had lived in prior to their initial migration from DIY.

The participants' last places of residence before moving to DIY were scattered across Indonesia. One participant's last place of residence was Dhahran, a city in Saudi Arabia. More than half of the participants had experienced repeat migration prior to returning to their place of origin. Notably, eight out of the 27 participants had migrated three or more times. During their migration period, participants maintained regular visits to DIY, regardless of the distance from their place of residence. Most participants returned to the province at least once a year, with some making multiple visits annually. These recurring visits facilitated the maintenance of social ties in their place of origin while enabling continued engagement with social networks in their destination communities. The demographic and socio-economic characteristics are detailed in the Table 1.

## 4.2 Adaptation strategies of retired older adult return migrants

As part of the process of migrants' adjustment to a social environment, adaptation requires the inclusion of individual migrants and the social environment [33]. Nevertheless, [31] argued that the adaptation process was not reciprocal between migrants and the destination community. Adaptation comprises a strong psycho-social transformation that necessitates migrants to adjust to a social environment distinct from their previous place of residence. This study refers to [31] to evaluate adaptation from an individual perspective. The subsequent section examines the strategies conducted by retired older adult return migrants to acclimate to the local population and social dynamics in DIY.

**4.2.1 Getting acquainted with the people living in the neighbourhood.** Getting acquainted with the residents of the new neighbourhood is a crucial process for migrants to familiarize themselves with the social life of the environment. This becomes increasingly important for individuals who have resettled in a neighbourhood that differs from their previous experiences in DIY. [33] suggested that this phase is essential in helping migrants understand and engage with the social life of their new neighbourhood. This study shows that some participants initiated the process of introducing themselves and notifying the neighbourhood about their move by visiting community leaders, such as the head of RT (*Rukun Tetangga* – the smallest group in a neighbourhood comprising minimum of 30–50 households) and the head of RW (*Rukun Warga* – a group consisting of multiple RTs). This approach fosters mutual acquaintance by enabling participants to become familiar with their neighbours. One participant elaborated on this process, stating the following:

> I introduced myself and my wife there (*the house of the head of RT and RW*), and chatted with them while registering, reporting, and informing them that we are new residents. After visiting the head of RT and RW, we visited our neighbours to get to know each other (**AY**, 63-year-old man).

Another strategy used to familiarize oneself with the social environment is extending an invitation to neighbours to visit the participant's house. This approach enables participants to establish connections with multiple neighbours simultaneously, as opposed to individually visiting each neighbour's home. One participant articulated this practice with the following statement:

> I moved here in 2012, and after we got here, I invited our neighbours, even at that time the house was not finished yet. It was no problem for us, the important thing was that we knew our neighbours as we lived here. We invited them to introduce ourselves. I then immediately looked for heads of *RT* and *RW* and stated that we would join the community (**PR**, 72-year-old man).

The participants' statements suggest that making friends and establishing familiarity with the surrounding neighbours is a crucial initial step in the adaptation process. This aligns with [29] that making friends is a facet of adaptation to a new community. Making friends has particular significance for individuals moving to an entirely unfamiliar area without prior residence.

**Table 1. Socio-demographic characteristics of the participants.**

| No. | Initials | Sex/ Age (year) | Place of birth (city/ regency, province) | Education | Previous place of residence (city/ district, province) | Previous occupation | Duration of leaving DIY before return migration (year) |
|---|---|---|---|---|---|---|---|
| 1 | WK | F/66 | Yogyakarta, DIY | Magister degree | Bekasi, West Java | Employee of a private national publisher | 35 |
| 2 | GB | M/72 | Temanggung, Central Java | Magister degree | Bekasi, West Java | Marketing director of a state-owned plantation company | 29 |
| 3 | SK | M/67 | Yogyakarta, DIY | Bachelor degree | Jakarta, DKI Jakarta | Employee of a foreign-owned company | 43 |
| 4 | EN | F/62 | Yogyakarta, DIY | Bachelor degree | Ambon, Maluku | Government official | 22 |
| 5 | NS | F/64 | Yogyakarta, DIY | Doctoral degree | Jakarta, DKI Jakarta | Government official | 33 |
| 6 | IM | M/71 | Bantul, DIY | Bachelor | Palu, Central Sulawesi | Government official (3rd rank) | 38 |
| 7 | SN | M/63 | Magelang, Central Java | Doctoral degree | Jakarta, DKI Jakarta | Government official (2nd rank) | 35 |
| 8 | MS | M/63 | Sleman, DIY | Bachelor degree | Batam, Riau Island | Government official (4th rank) | 39 |
| 9 | TW | M/68 | Yogyakarta, DIY | Bachelor degree | Gresik, East Java | Production manager of a state-owned enterprise | 30 |
| 10 | BS | M/66 | Malang, East Java | Bachelor degree | Bontang, East Kalimantan | HRD manager of a state-owned enterprise | 28 |
| 11 | AY | M/63 | Purbalingga, Central Java | Bachelor degree | Mandau, Riau | Plant Heavy Oil Operation Manager of a foreign-owned company | 29 |
| 12 | SU | F/67 | Yogyakarta, DIY | Bachelor degree | Jakarta, DKI Jakarta | General Manager of Affiliated Holding Companies Division of a state-owned enterprise | 39 |
| 13 | SL | M/62 | Gunung Kidul, DIY | Bachelor degree | Ternate. North Maluku | Government official (3rd rank) | 37 |
| 14 | SY | M/66 | Gunung Kidul, DIY | Bachelor degree | Kudus, Central Java | Government official (principal of a state junior high school) | 39 |
| 15 | HD | M/61 | Kulon Progo, DIY | Bachelor degree | Serang, Banten | Government official (4th rank) | 25 |
| 16 | SW | M/68 | Kulon Progo, DIY | Diploma 3 degree | Jakarta, DKI Jakarta | Government official (staff) | 38 |
| 17 | BB | M/65 | Cilacap, Central Java | Bachelor degree | Jakarta, DKI Jakarta | Production manager of a private-owned oil palm plantation | 35 |
| 18 | HS | M/60 | Purworejo, Central Java | Bachelor degree | Jakarta, DKI Jakarta | Commissioner board of a state-owned enterprise | 30 |
| 19 | SD | M/72 | Yogyakarta, DIY | Master degree | Bandung, West Java | Lecturer at a state university | 47 |
| 20 | SM | F/66 | Yogyakarta, DIY | Master degree | Jakarta, DKI Jakarta | Researcher at a state research institution | 39 |
| 21 | AT | F/62 | Purwokerto, Central Java | Bachelor degree | Bontang, East Kalimantan | Senior high school teacher | 24 |
| 22 | PR | M/72 | Sleman, DIY | Diploma 3 degree | Palembang, South Sumatra | Principal of a Catholic junior high school | 42 |
| 23 | MM | M/65 | Ambon, Maluku | Bachelor degree | Dhahran, Saudi Arabia | Senior Development Geologist of a Saudi Arabia-owned oil company | 32 |
| 24 | SH | F/62 | Kulon Progo, DIY | Senior high school degree | Wonosobo, Central Java | Government official (staff) | 37 |
| 25 | TH | M/65 | Kulon Progo, DIY | Bachelor degree | Jakarta, DKI Jakarta | Senior high school teacher | 40 |
| 26 | KM | F/64 | Kulon Progo, DIY | Bachelor degree | Jakarta, DKI Jakarta | Principal of a state elementary school | 39 |
| 27 | HY | M/64 | Madiun, East Java | Bachelor degree | Jakarta, DKI Jakarta | Government official (4th rank) | 36 |

**4.2.2 Shedding socio-economic attributes before retirement.** Participants acquired specific socio-economic attributes prior to retirement. Some attained high-ranking positions in their respective offices, wielding influence over the management and maintenance of office performance. Additionally, a few participants hold positions of honour in their former places of residence. To successfully adapt to a new community, these individuals deliberately shed their attributes and present as ordinary members of the neighbourhood. The following is a quotation from a participant who held a high-ranking government position before retirement:

> Even though I had a doctoral degree, I had a high position at the former office, but in the community, I try not to show it. In the past, I often made speeches everywhere (*laugh*). When the audience did not listen to my speech, I got angry. When I made a speech and there were some members who chatted with each other, I got angry. Now I have to place myself in the opposite position (**SN**, 67-year-old man).

Relinquishing socio-economic attributes ascribed to the participants was a deliberate choice to refrain from showing the pre-retirement status. This is an attempt to remain humble and conformist towards others and thus more easily accepted by people, which in turn mitigates barriers in the adaptation process.

**4.2.3 Appreciating and respecting others in the neighbourhood.** Appreciating and respecting people living in the neighbourhood is a strategy used by participants to adapt to the social environment. The study shows that participants made some attempts to appreciate and respect individuals residing in the same neighbourhood as a means of adapting to the social environment. The first is the use of polite and appropriate language according to the age and social status of interlocutors. One participant articulated the communication practices with people in the vicinity as follows:

> I grew up in a Javanese family that instilled the polite manners of language. The Javanese language is divided into different levels of language. When I came here, I knew which one (level of language) to use with whom, which one with the others. Therefore, there is no feeling of being un-respected among my interlocutors. For example, when I speak *ngoko* (*ngoko* is the lowest level of the Javanese language) with anyone, people feel un-respected. I speak *boso* (*boso* is a higher level of Javanese language) to anyone who is not very close to me. I know exactly which language level to use and which language level for whom. Language is very helpful to get along with others (**WK**, 66-year-old woman).

In Javanese society, the use of appropriate language for the interlocutor's age and other social attributes is a crucial aspect of communication. The practice promotes a sense of respect to acknowledge individuals, and this contributes to the local community's openness in accepting retired older adult return migrants into the neighbourhood.

The second effort made by participants to adapt to their neighbourhoods includes maintaining a quiet attitude and giving in to the behaviour of others. This behaviour reflects a form of tolerance aimed at preventing conflicts, particularly with the present younger generation, whose attitudes differ from the previous generation. Most participants show acceptance of the younger generation's attitudes and refrain from expressing negative responses. This is a deliberate effort to uphold harmony in the social environment, and one participant stated:

> Like me, for example, in that corner, that terraced house, there are many boarders. At certain times, the house was crowded and the boarders shouted and made noise. I have to adjust my way of dealing with the millennial generation. For me, as long as they did not break the law and did not bother me, it would be no problem. There has been a notable social change; in the past, boarders adhered to local social norms and conformed to them, whereas current boarders no longer follow that pattern (**SN**, 67-year-old man).

The third attempt is the sincere acceptance of local people, with all traditions and customs, which represents another manifestation of appreciating and respecting others. This attitude indicates a lack of superiority complex, showing a

genuine intention to integrate with others in the community. The approach significantly contributes to promoting a harmonious social life in the new destination. This was stated by one participant as follows:

> This is like a proverb, "*di mana bumi dipijak, di situ langit dijunjung*" (literally means where the earth is stepped, there the sky is upheld). Therefore, we have to follow what local people practice and there is no difficulty because I have ever lived in Yogya and my wife is from DIY (**MM**, 67-year-old man).

The efforts made show that the participants accepted and respected people with all their conditions. This facilitates the integration into the social fabric of the neighbourhood, promoting a sense of community that significantly eases assimilation. [82] argued that assimilation and integration are predominant aspects of adaptation. In terms of language usage, these results are in line with the study of [37] in Australia. The study revealed that assimilation of migrants with the native-born population in terms of English fluency is identified as a contributing factor in promoting adaptation to the new social life in the country. In the context of this study, fluency in English can be analogized to the ability to use the Javanese language that is appropriate for the interlocutor.

**4.2.4 Participating in various social community activities.** Relocating to a new destination includes the disruption of social networks and familial relationships in the community where the migrants previously resided [39]. Therefore, a new social connection is established by actively participating in various local groups. Engaging in community social activities is a deliberate strategy used to assimilate and integrate seamlessly with residents. This active inclusion promotes a sense of belonging, facilitating a smoother adaptation process. One participant, who held the position of branch manager in a private-owned oil palm plantation prior to retirement, articulated the perspective as follows:

> Once a month, I conduct a get-together with other people in the neighbourhood. Even here I join the *ronda* (activity carried out by adult men, usually head of household, who gather at night to guard the neighbourhood of crimes and other harmful activities). I join the *ronda*, walking around the neighbourhood at night. The activity is more likely to be get-togethers with neighbours around my house (**BB**, 65-year-old man).

Participating in community social activities is not solely an attempt made by individuals who relocate to different neighbourhoods upon returning to DIY. Individuals who return to their former neighbourhoods also engage in community social activities to facilitate integration into the current social dynamics. One participant expressed that residing outside the village for nearly four decades has rendered her unfamiliar with the local social life. Therefore, she participated in many social and religious activities to re-adapt and get attached to the social life of the community. The following is a quotation from her perspective.

> Although I often go back to my hometown during my time living outside the area, I felt like a newcomer at the time I returned here. I have to learn more about social life, and I mostly participated in recitation, aerobics exercises, and social gatherings to learn and familiarise myself with the community (**SH**, 62-year-old woman).

**4.2.5 Join groups that match with interests and hobbies.** This study shows that some participants actively engage with groups in line with their hobbies and interests. The groups may not be close to their residences, but are located in the same city or district. This deliberate choice serves as a strategic approach to cultivating friendships beyond the immediate neighbourhood, thereby promoting a smoother adaptation to the new social environment. One participant, who held a managerial position in a state-owned enterprise prior to retirement, reported this insight in the following statement:

> The first thing I did here I look for a community. Initially, I looked for a table tennis community, I looked for a coach, and I joined the club because I was like an ordinary person since I am not a high-ranking officer anymore (**GB**, 72-year-old man).

Making friends in a new social environment is one aspect of social life that is included in the adaptation process [29]. Participating in a group can serve as an effort to connect with individuals and cultivate friendships. Engaging in activities and sharing time with fellow members plays a significant role in assisting retired older adult return migrants to integrate with the local community.

## 5 Discussion and conclusion

Migration led to a change of residence and a corresponding alteration in the social environment. Therefore, migrants were required to acclimate to the social dynamics of the new area. Return migrants were subjected to an adaptation process for readjusting to social life in the area. This adaptation was necessary due to alterations in values resulting from migration experiences and potential changes in the social dynamics of the area of origin.

As expressed in the statements, retired older adult return migrants who participated in this study experienced a smooth adaptation process in adjusting to the social environment after returning to DIY during the retirement phase. The ease of adaptation can be attributed to a continued embrace and practice of Javanese cultural values, which serve as a guide for navigating social life and integrating into the community in their original area. The participants remained deeply ingrained with Javanese cultural values despite living outside DIY for extended periods of 20–40 years. The Javanese cultural values of *gotong royong* (helping one another as community members), *grapyak semanak* (friendly and easy to get along with others), *lembah manah* (humble), *ewuh pakewuh* (respect the feelings of others), and *tepo seliro* (tolerance) persisted strongly among retired older adult return migrants. This inherent adherence to Javanese cultural values significantly aided the migrant in navigating the adaptation process with greater ease.

The participants shed socio-economic attributes acquired in place of residence before moving back to DIY is a form of commitment to the Javanese cultural value of *lembah manah* (humility), while abstaining from flaunting socio-economic standing. This is an endeavor to stay humble and fit in with the people in the neighbourhood. This aligns with the social principles of neighbourhood life and group membership in Javanese society, which emphasize maintaining conformity with others and avoiding the attainment of individual prominence within the group [75,83].

The use of Javanese language appropriate to interlocutors is a commitment to the Javanese cultural values of *ewuh pakewuh* or *pangerten*, meaning respect the feelings of others and *andhap ashor* (respect others who are older or superior). This practice plays an important role in conducting social interactions in the DIY region. Expressions of mutual respect are also manifested in other forms, including accepting others with all their conditions. The participants' acceptance of the attitudes of the young generation who are different from the previous generation is a manifestation of *tepo seliro*, a value in Javanese culture signifying tolerance and understanding towards others. The ability to accept the attitudes of the younger people can be shown by recognizing differences and coexisting harmoniously in society.

The result of the study aligns with research on international migrants' adaptation in their destination countries, where language proficiency has been identified as a crucial factor supporting the adaptation process [37,49]. The significance of language acquisition in the adaptation process was also highlighted by [30] in a study examining the experiences of French international students at Thai universities. Unlike these studies, which investigate migration to locations with distinct linguistic environments, the present study on retired older adults concentrates on the adaptation of return migrants interacting with residents who share the same native language. Proficiency in using various levels of the Javanese language appropriately may be analogous to fluency in a foreign language in other studies [30,37,49]. Hence, the ability to navigate linguistic nuances in the shared language becomes a significant factor in the adaptation process for retired older adult return migrants.

In terms of participation in community activities, this study shows a similar phenomenon to a study conducted by [84] in the United Kingdom. The study on pre-retirement age migration to three rural areas in the country reported that migrant household heads engaged in both social and special interest activities, with the highest levels of participation observed among the migrant population. Participating in the social and religious groups in the community enables migrants to

engage with others [39]. This inclusion develops a sense of belonging in the groups to promote mutual acceptance and a smoother adaptation process for migrants.

Indonesia comprises more than thirty provinces inhabited by ethnically diverse populations with distinct cultural values, some of which are shared across ethnic groups. The shared values facilitate intercultural adaptation, particularly in migration destination areas. This adaptation process is exemplified by the Minangkabau people, a major ethnic group originating from West Sumatra Province. Minangkabau migrants often integrate cultural principles into the social fabric of new communities in entirely unfamiliar destinations or after returning to their place of origin. An example of the principle is embodied in the Minangkabau proverb *Aia urang disauak, rantiang urang dipatah, adaik urang dituruik*, which translates roughly as "follow the customs of the place you go". This philosophy emphasizes the importance of humility and culturally appropriate behavior, promoting migrants' conformity to local norms [85].

Other Minangkabau cultural values, such as *timbang raso* (tolerance) and *sifat kesayangan* (affectionate or loving traits), emphasize the importance of respecting others through courteous or polite communication [86]. Research on the adaptation of Minangkabau migrant students in Jakarta reported that communication style was adjusted in line with local norms to promote greater social closeness and acceptance [87]. These cultural values play a crucial role in facilitating migrants' adaptation to the social environment in the destination communities when manifested in attitudes and behaviours during social interactions.

The adaptation of the Bugis migrants, an ethnic group in South Sulawesi Province, is strongly influenced by the cultural values. An example of these values is *sipakatau*, which emphasizes mutual respect and the recognition of shared humanity [88]. Individuals are expected to appreciate, respect, and care for others regardless of social, cultural, or ethnic background by adhering to *sipakatau* [89]. In the context of migration, this value plays a significant role in guiding interpersonal interaction within destination communities, facilitating smoother social adaptation for Bugis migrants.

Based on [26]'s model of acculturation, the retired older adult return migrants in this study adopted affirmative and intercultural strategies upon returning to their place of origin. Both strategies show the participants' eagerness to maintain the culture of origin while living outside DIY during the migration period. The participants maintain their home culture by paying regular visits to family and relatives living in DIY. Participants who were not born in DIY and had no relatives residing in the area made repeated visits during holidays. They also maintained cultural ties to their former migration destinations outside DIY by keeping in contact with long-standing friends from those regions.

The two strategies support the retired elderly in adaptation and prevent them from distress while living back in their original area. Therefore, these migrants expressed contentment and a desire to remain in the area with no intention of migrating again. These findings contrast with [34] in which some older adult return migrants to Turkey preferred to return to their previous countries of residence.

The older adult participants represented diverse socio-demographic and economic backgrounds, but the adaptation strategies were remarkably similar. This uniformity is primarily attributed to adherence to Javanese cultural values maintained despite having left the place of origin long ago. These values are deeply embedded in the social fabric of Yogyakarta and are embraced by residents regardless of their socio-demographic or economic status. Furthermore, individuals who are not ethnically Javanese but have previously lived in the Special Region of Yogyakarta tend to adopt these cultural norms. These shared norms facilitate social adaptation among retired older adult returnees.

## 6 Policy recommendation

Indonesia is experiencing a demographic transition marked by population aging, with some provinces meeting the criteria for an aging population. The older adult population residing in the provinces consists of both individuals aging in place and older adult migrants who have relocated with the intention of spending their later years in the regions. A province receiving a significant number of older adult migrants is the Yogyakarta Special Region (DIY).

Continued return of older adult retirees to DIY influences the local government in designing programs for the population, particularly basic services for older people. DIY has 16.6% of persons 60 years and older, the highest among other provinces in Indonesia [90]. This percentage comprises non-migrant and migrant individuals, including those who returned to DIY after retirement. Since DIY is considered a favourable place to spend later years, the percentage of older adults increases more rapidly than in other provinces. This inference is drawn from the findings of this study, where retired older adult return migrants feel at home in the province with no desire to relocate.

Other provinces serving as destinations for retired older adults may encounter similar challenges. However, these may be less pronounced than those faced by the Special Region of Yogyakarta (DIY). As the trend of return migration among retired older adults continues [15,19–22], many provinces with historically high rates of outmigration may experience an influx of this demographic. Therefore, local governments will bear increased responsibility for addressing the needs of older adults who have aged in place and returned to their home provinces after retirement.

Older adults require special attention due to declining physical abilities and age-related health conditions. The number of older individuals is directly proportional to the demand for services provided. Therefore, the local government of the Special Region of Yogyakarta offers more services related to older adults compared to other regions. The DIY government must ensure the availability of adequate health services and infrastructure to support the needs of the ageing population. In addition, the local governments should be able to increase community participation to meet the needs of this population. Healthy and physically capable older adults can serve as drivers of community activities. The individuals can actively participate in various activities suitable for their conditions regularly, when organized in groups, such as exercise and recreational activities. This proactive inclusion contributes to reducing the government's burden in meeting the needs of the demographic group.

Apart from providing essential services for older adults, local governments, in collaboration with community members, are also responsible for preserving Javanese cultural values, particularly among younger generations. Technological advancements, such as the widespread availability of internet connectivity in various forms, have significantly influenced societal values, attitudes, and behaviours. Increased engagement in online communication, especially among youth, has contributed to a decline in face-to-face interactions. Moreover, the nature of internet-based communication often differs from traditional modes of interaction and may be perceived by older individuals as impolite or disrespectful [91]. A significant challenge of online communication is its potential to reduce social interaction [92], which can consequently impact interpersonal behaviour. Specifically, the absence of face-to-face contact complicates the demonstration of respect, particularly in interactions with older adults. This difficulty may hinder older people's adaptation to contemporary social dynamics, especially concerning younger generations accustomed to digital communication.

## Author contributions

**Conceptualization:** Mita Noveria, Umi Listyaningsih, Agus Joko Pitoyo, Aswatini Anaf.

**Data curation:** Mita Noveria.

**Formal analysis:** Mita Noveria, Umi Listyaningsih, Agus Joko Pitoyo, Aswatini Anaf.

**Funding acquisition:** Mita Noveria, Aswatini Anaf.

**Investigation:** Mita Noveria, Umi Listyaningsih, Agus Joko Pitoyo, Aswatini Anaf.

**Methodology:** Mita Noveria, Umi Listyaningsih, Agus Joko Pitoyo, Aswatini Anaf.

**Project administration:** Mita Noveria.

**Resources:** Mita Noveria, Umi Listyaningsih.

**Supervision:** Mita Noveria, Umi Listyaningsih, Agus Joko Pitoyo, Aswatini Anaf.

**Validation:** Mita Noveria, Umi Listyaningsih, Agus Joko Pitoyo, Aswatini Anaf.

**Visualization:** Mita Noveria.

**Writing – original draft:** Mita Noveria.

**Writing – review & editing:** Mita Noveria, Umi Listyaningsih, Agus Joko Pitoyo, Aswatini Anaf.

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
