## [Decision Letter · Decision Letter 0]

20 Apr 2025

Dear Dr. Noveria, 

Thank you for submitting your manuscript to PLOS ONE. After careful consideration, we feel that it has merit but does not fully meet PLOS ONE’s publication criteria as it currently stands. Therefore, we invite you to submit a revised version of the manuscript that addresses the points raised during the review process.

Please make sure to address all the comments of the two reviewers, and have the revised manuscript edited for English. 

We look forward to receiving your revised manuscript.

Kind regards,

Gouranga Lal Dasvarma, PhD

Academic Editor

PLOS ONE

Journal Requirements:

 [The research was partly funded by the Institute of Social Sciences and Humanities of the National Research and Innovation Agency, Indonesia (Grant number B-577/III.7/HM.06.02/2/2023).].

 [The research was partly funded by the Institute of Social Sciences and Humanities of the National Research and Innovation Agency, Indonesia (Grant number B-577/III.7/HM.06.02/2/2023).]. 

6. Please include a separate caption for each table in your manuscript.

Additional Editor Comments:

Please address all the comments of the two reviewers and have the revised manuscript edited for English.

Reviewers' comments:

Reviewer's Responses to Questions

**Comments to the Author**

1. Is the manuscript technically sound, and do the data support the conclusions?

Reviewer #1: Yes

Reviewer #2: Yes

2. Has the statistical analysis been performed appropriately and rigorously?

Reviewer #1: Yes

Reviewer #2: N/A

3. Have the authors made all data underlying the findings in their manuscript fully available?

Reviewer #1: No

Reviewer #2: No

4. Is the manuscript presented in an intelligible fashion and written in standard English?

Reviewer #1: Yes

Reviewer #2: Yes

Reviewer #1: Your topic is important because in your words “such studies of this age-group are scarce”. However, I offer the following recommendation about the way in which you frame your study. You say it is about “retired older adult return migrants’ adaptation strategies who migrate to their place of origin”. I suggest this characterisation needs to be framed more definitively as a study of well-educated and high-achiever retirees’ adaptation strategies on return to Yogyakarta (DIY) as their place of origin.

I recommend this for two reasons: 1) the sample population is over-represented in the Indonesian education achievement data. While the participants’ educational attainment varied from senior high to tertiary school graduates; the country’s education data shows (as of 2023) just around 40% of Indonesians aged 15 years and above had completed senior high school or more. 2) In your words, “DIY is considered an ideal residential destination for older adults” (p.12.) My comments and recommendations aim 1) to deflect the potential for readers to incorrectly generalise from the research data; and 2) to apply context specificity to questions you raise about social issues related to mal-adaptation, i.e. DIY as “an ideal location” encompasses specific consequences of mal-adaptation associated with its custodianship of customary values. But on this question, it would greatly benefit the text if you specified which social issues “potentially disrupting the harmony of a social environment”.

I also suggest that you add a section on policy recommendations. Your study produces sound evidence on the necessity for the provision of services to achieve well-being in old age. While the DIY location necessitates discussion of social protection relevant to DIY, such recommendations could also be useful to other regions beyond the study location given the importance of housing, social protection, age discrimination, and country-wide social valuation of mutuality.

On grammar, the following are two noticed instances where minor corrections are needed: 1) Studies on the adaptation of older adults who migrate internally, particularly to their place of origin, have not been conducted in Indonesia, despite this group of population contribute to the number of return migrants (p.5.)

2) The Yogyakarta Special Region Province is a central of Javanese culture, apart from Solo, a city in Central Java Province (p.5).

Reviewer #2: 6 Review Comments to the Author (minimum 200 characters)

This study focuses on the adaptation strategies of retired older adult return migrants. While many theories and studies focus on migration, this research pays attention to return migration in relation to a specific age cohort, older adults, and the strategies they follow to adapt to the socio-cultural environment of the region they left sometimes back. Therefore, the paper brings new knowledge to academia.

Abstract

Include the recommendations at the end of the abstract.

Introduction

Previous literature has been incorporated sufficiently. The researchers have reviewed relatively recent studies related to the theme including theories/models. The research gap was identified to make the baseline of the study.

Methodology

Authors employ qualitative approach to investigate the adaptation strategies of the retired older adult return migrants. The objective could be 'to examine the adaptation of retired older adult return migrants in their place of origin.' It focuses on the strategies used to socially integrate into the community with a particular emphasis on the neighbourhood. The findings of the research are based on 27 in-depth interviews using snowball technique.

Research questions do not appear in the paper. Include the research questions.

Results

At the beginning of the results section, authors presented the socio-economic characteristics of the respondents both as a description and as a table, repeating the information. It is better to analyze the characteristics by gender, age, previous occupation, duration of the previous residence, etc., and rearrange the explanation on pages 13-14.

The authors have identified 5 adaptation strategies of the respondents. However, the extent of adaptation may vary by gender, education level, previous occupation and so on. It is suggested to discuss the adaptation strategies according to these variables.

-The main weakness of the paper is repetition, particularly in introduction and literature review.

-The final manuscript should be edited by a professional language editor.

References:

In-text citations - need to be corrected in one place on page 21.

**Do you want your identity to be public for this peer review?** For information about this choice, including consent withdrawal, please see our Privacy Policy

Reviewer #1: **Yes: ** Dr James Chalmers, Flinders University, Australia

Reviewer #2: No

---

## [Author Response · Author response to Decision Letter 1]

26 May 2025

I have responded to editor's and reviewers' comments in the two files:

1. The file of "Cover Letter" consists of responses to academic editor's comments regarding research funding, roles of the funder, and data accessibility.

2. The file of "Response to Reviewer" consists of responses to the two reviewers regarding of revision of some parts of the manuscript as required.

---

## [Editor Report · Decision Letter 1]

6 Jun 2025

Dear Dr. Noveria ,

Thank you for submitting your manuscript to PLOS ONE. After careful consideration, we feel that it has merit but does not fully meet PLOS ONE’s publication criteria as it currently stands. Therefore, we invite you to submit a revised version of the manuscript that addresses the points raised during the review process.

We look forward to receiving your revised manuscript.

Kind regards,

Gouranga Lal Dasvarma, PhD

Academic Editor

PLOS ONE

Journal Requirements:

**Additional Editor Comments: **

**It appears that you have addressed all the comments from the two reviewers. However, errors in grammar and expression still remain in the manuscript. In particular I find the title a little awkward. I suggest you rite the title as "Adaptation of retired older adult return migrants in place of origin in Yogyakarta Special Region, Indonesia" . Yogyakarta is a Special Region (Daerah Istimewa), not a province. You can add a footnote at the appropriate place, for example: "Administratively, Indonesia is divided into 30 provinces and two special regions, the latter having special autonomous status, with greater control over certain aspects of governance than a regular province". Further, please have the entire manuscript thoroughly edited for English by a professional editor.**

---

## [Author Response · Author response to Decision Letter 2]

24 Jun 2025

I have responded reviewers and academic editor comments, as follow:

1. I agree with the comments of both reviewers and I have revised the manuscript in accordance to their comments.

2. I accepted academic editor's suggestion on slightly change of manuscript's title and I have change the title.

3. Reviewer#2 and academic editor suggested the entire manuscript thoroughly edited for English by a professional editor.

It has been edited for English by a professional editor (Cambridge Proofreading) and I attached certificate of English editing on response to reviewers and cover letter files.

---

## [Editor Report · Decision Letter 2]

26 Jun 2025

Adaptation of retired older adult return migrants in their place of origin in the Yogyakarta Special Region, Indonesia

PONE-D-24-29661R2

Dear Dr. Mita Noveria, 

We’re pleased to inform you that your manuscript has been judged scientifically suitable for publication and will be formally accepted for publication once it meets all outstanding technical requirements.

Kind regards,

Gouranga Lal Dasvarma, PhD

Academic Editor

PLOS ONE

Additional Editor Comments (optional):

Thank you for carrying out the amendments.
---

## [Editor Report · Acceptance letter]

PONE-D-24-29661R2

PLOS ONE

Dear Dr. Noveria,

I'm pleased to inform you that your manuscript has been deemed suitable for publication in PLOS ONE. Congratulations! Your manuscript is now being handed over to our production team.

Kind regards,

on behalf of

Dr. Gouranga Lal Dasvarma

Academic Editor

PLOS ONE